# Moving towards a strategy to accelerate cervical cancer elimination in a high-burden city—Lessons learned from the Amazon city of Manaus, Brazil

**Kátia Luz Torres**[1,2,3☯]*, **Heidy Halanna de Melo Farah Rondon**[2,3☯], **Toni Ricardo Martins**[4], **Sandro Martins**[3], **Ana Ribeiro**[3,5], **Taina Raiol**[3], **Carla Pintas Marques**[3,6], **Flavia Corrêa**[7], **Arn Migowski**[7], **Thais Tâmara Castro e Minuzzi-Souza**[3,8], **Mark Schiffman**[9], **Ana Cecilia Rodriguez**[9], **Julia C. Gage**[9]

**1** Amazon State Oncology Control Foundation (FCECON), Manaus, Amazonas, Brazil, **2** Post Graduation Program in Health Sciences and Basic and Applied Immunology at The Federal University of Amazonas (UFAM), Manaus, Amazonas, Brazil, **3** Center for Epidemiology and Health Surveillance, Oswaldo Cruz Foundation (Fiocruz), Brasília, Federal District, Brazil, **4** Tropical Medicine Institute, São Paulo University, Virology Laboratory (LIM52) (USP-SP) - São Paulo, São Paulo, Brazil, **5** Department of Pharmacy, Faculty of Health Sciences, University of Brasília (UNB), Brasília, Federal District, Brazil, **6** Collective Health, Universidade de Brasília (UnB), Brasília, Federal District, Brazil, **7** Cancer Early Detection Division, Brazilian National Cancer Institute (INCA), Rio de Janeiro, Rio de Janeiro, Brazil, **8** National Immunization Program, Health Surveillance Secretariat (SVS), Ministry of Health, Brasília, Federal District, Brazil, **9** Division of Cancer Epidemiology and Genetics, National Cancer Institute, NIH, Bethesda, Maryland, United States of America

☯ These authors contributed equally to this work.
* katialuztorres@hotmail.com

## Abstract

The World Health Organization Call to Eliminate Cervical Cancer resonates in cities like Manaus, Brazil, where the burden is among the world's highest. Manaus has offered free cytology-based screening since 1990 and HPV immunization since 2013, but the public system is constrained by many challenges and performance is not well-defined. We obtained cervical cancer prevention activities within Manaus public health records for 2019 to evaluate immunization and screening coverage, screening by region and neighborhood, and the annual Pink October screening campaign. We estimated that among girls and boys age 14–18, 85.9% and 64.9% had 1+ doses of HPV vaccine, higher than rates for age 9–13 (73.4% and 43.3%, respectively). Of the 90,209 cytology tests performed, 24.9% were outside the target age and the remaining 72,230 corresponded to 40.1% of the target population (one-third of women age 25–64). The East zone had highest screening coverage (49.1%), highest high-grade cytology rate (2.5%) and lowest estimated cancers (38.1/100,000) compared with the South zone (32.9%, 1.8% and 48.5/100,000, respectively). Largest neighborhoods had fewer per capita screening locations, resulting in lower coverage. During October, some clinics successfully achieved higher screening volumes and high-grade cytology rates (up to 15.4%). Although we found evidence of some follow-up within 10 months post-screening for 51/70 women (72.9%) with high-grade or worse cytology, only 18 had complete work-up confirmed. Manaus has successfully initiated HPV vaccination, forecasting substantial

**Data Availability Statement:** All relevant data are within the paper and its Supporting information files.

**Funding:** The funders had no role in study design, data collection and analysis, decision to publish, or preparation of the manuscript.

**Competing interests:** The authors have declared that no competing interests exist.

cervical cancer reductions by 2050. With concerted efforts during campaigns, some clinics improved screening coverage and reached high-risk women. Screening campaigns in community locations in high-risk neighborhoods using self-collected HPV testing can achieve widespread coverage. Simplifying triage and treatment with fewer visits closer to communities would greatly improve follow-up and program effectiveness. Achieving WHO Cervical Cancer Elimination goals in high-burden cities will require major reforms for screening and simpler follow-up and treatment.

## Introduction

Cervical cancer has a well-described natural history with a long sojourn allowing for prevention with proper screening and follow-up including simple treatment. This cancer also has an identified causal agent–persistent infection with one or more of approximately 12 high-risk genotypes of the Human Papillomavirus (HPV) [1, 2]–that has created an opportunity to develop better screening methods and vaccination against the most carcinogenic types. Yet, the burden of cervical cancer cases remains a formidable public health challenge with high incidence and mortality rates in many countries, including Brazil [3, 4]. In 2020, Brazil expected 15.38 new cervical cancer cases per 100,000 women, with a much higher incidence in the Amazon State and Manaus (40.18 and 61.54 cancer cases per 100,000 women, respectively) [5]. These elevated rates persist in spite of offering free cytology-based cervical cancer screening since 1990. The Amazon region is characterized by social, geospatial and environmental challenges to implement cervical cancer screening including seasonal river floods and droughts, difficulty accessing health services, low educational level and early sexual initiation [6–8].

The quadrivalent HPV vaccination was introduced in Amazonas as a girls-only school-based program in 2013 as a local initiative and extended to all Brazil in 2014 [9]. The program consisted of two doses for girls age 9 to 14. In 2015, it was modified to be offered no longer in schools, but in public and private health units. Boys age 11 to 14 were added in 2017 [10].

Cervical cancer screening in Manaus, as throughout the rest of the country, is a cytology-based program performed at decentralized Basic Health Units (Unidades Básicas de Saúde, UBS), and Brazilian health program that includes a multiprofessional team called Family Health Program (that includes a primary care assistance or referral), five colposcopy clinics and reference treatment centers. All these services are offered by a public health system (SUS in the Portuguese acronym). It follows the 2016 Brazilian National Cervical Cancer Screening Guidelines that recommend cytology testing every 3 years after 2 consecutive negative results for women aged 25 to 64. Women with abnormal findings require either repeat cytology in 6 or 12 months or referral to colposcopy with biopsy and excisional treatment as needed [11]. Annual community campaigns like "Pink October" or more recently "Purple March" are utilized to increase coverage.

To better understand the reasons for the sustained high cervical cancer burden in Manaus we analyzed vaccination coverage plus two critical program components identified in Brazilian National Cervical Cancer Screening Guidelines [11]: screening coverage, follow-up colposcopy and treatment, during 2019. We aim to inform potential programmatic areas of improvement for the cervical cancer prevention program in Manaus and other similar screening programs world-wide.

## Materials and methods

Data pertaining to cervical cancer prevention activities within the Manaus SUS public health records were obtained for 2019 and 2020. Annual numbers of vaccines were extracted from the National Immunization Program Information System (SI-PNI) [12]. Population estimates were collected from the Brazilian Geographic and Statistical Institute [13]. The HPV vaccine coverage indicator was calculated using the approach described by the Pan American Health Organization [14]. Specifically, the total number of girls age 9–18 and boys age 10–15 residing in Manaus in 2019, was divided by the estimated number of girls and boys in that cohort ever receiving one or two doses between 2013 and 2019.

The number of cytologies processed were extracted from SISCOLO (the cervical cancer information system that was utilized in Manaus) and results were obtained from the only centralized local public cytopathology laboratory. Screening coverage in Brazil is monitored using the metric percent of the target population screened with the goal of 80% coverage of 1/3 of the target population each year [11]. This was calculated by dividing the annual number of cytopathological exams collected among women age 25 to 64, by the target number of women to be screened that year—estimated population of women age 25–64 divided by 3—the recommended screening interval. With cooperation from the city´s women's health control program, screening coverage estimates were also calculated from 2010 through 2019 with data from SISCOLO and Information Management and Health Situation Analysis Division of Manaus Secretary of Health (DICAR/SUBGS/SEMSA/MANAUS) [15]. Calculations of coverage assume that women are screened at the recommended intervals. Our 2019 calculations of coverage considered only satisfactory cytology slides performed exclusively for screening, excluding cytologies performed for clinical follow-up.

To identify whether women at high risk attended screening in 2019, the proportion of cytology with high-grade squamous intraepithelial lesion or worse (HSIL$^+$) results was calculated [16]. We included HSIL as well as atypical squamous cells cannot rule-out high-grade squamous intraepithelial lesion (ASC-H), atypical glandular cells (AGC), and squamous cell carcinoma (SCC).

The referral appointment system (SISREG in the Portuguese acronym) for public health services was used to identify cervical cancer cases referred from July-December 2019 to the city's public cancer reference hospital, Fundação Centro de Controle de Oncologia do Estado do Amazonas (FCECON). We used the residential address reported at time of diagnosis or treatment to determine if the patient was covered by the Manaus program and assign her residential zone. We stratified our analysis by residential zone (4 urban and 1 rural shown in S1 Fig) and used estimations of population to calculate and compare the indicators of screening coverage, HSIL$^+$ and estimated 2019 cancer incidence for each urban zone. Differences by zone were calculated using Chi-square tests of trend or 95% confidence intervals.

The population size, number of screening units offering cytology (includes family health teams and UBS centers), and screening coverage for each neighborhood of Manaus was obtained from DICAR/SEMSA. For each neighborhood, the estimated population size, number of women age 25–64 per screening unit and screening coverage were compared. Pearson's correlation coefficient was used to compare women per screening unit and screening coverage.

We selected 12 UBS clinics with a history of high screening volume and adequate data management to evaluate HSIL$^+$ rates and follow up during the Pink October campaign in 2019. We used secondary data from SISREG and SISCOLO to identify the extent to which women with HSIL$^+$ screening results received complete follow-up. SUS colposcopy teams provided further information from clinic records and management reports. Follow-up activities conducted through August 2020 were included in this analysis.

This research protocol was approved by the National Research Ethics Commission–CONEP/Brazil under number N. 3.676.252. Data were collected in collaboration with the local health authorities and the IRB waived the requirement for informed consent. For the analysis, all data were fully anonymized.

## Results

Manaus is the capital city of Amazonas State in northern Brazil with a total area of 11,401 km$^2$, most of which is rural (96.3%) (S1 Fig). Manaus is considered the seventh most populated city in Brazil with approximately 2,182,763 inhabitants in 2019, the majority of whom live in the urban zones (99.5%). Selected geographic, demographic, and health system characteristics are described in S1 Table. HPV vaccination coverage in 2019 for girls age 9–13 was 73.4% with one dose and 47.4% with two doses and for boys age 10–13 was 43.3% and 22.2% respectively. Coverage was higher for girls age 14–18 (85.9% and 71.3%, first and second doses, respectively) and for boys age 14–15 (64.9% and 33.3%, first and second doses, respectively) (Table 1).

In 2019, 72,230 screening cytology tests were collected among women age 25–64, corresponding to 40.1% of the target population. A similar coverage rate has been observed for the past 10 years with the highest coverage registered in 2017, 51.1% (Fig 1). Notably, an additional 17,979 screening cytology tests were collected among women outside the target age group, representing 24.9% of the screening tests performed in 2019. The proportion of cytologies with HSIL$^+$ results was 2.1% with 0.3% representing glandular lesions (AGUS and AGC).

The regional analyses of Manaus compared outcomes between the city's 4 urban zones (East, North, West, South). Table 2 shows, for each of these zones, the estimated female population aged 25–64, number of cytologies collected, screening coverage and the proportion of HSIL$^+$ and cancer cases. Screening coverage was highest in the East zone (49.1%) and lowest in the South and West zones (32.6% and 33.3%, respectively, p<.0001 for both comparisons). The HSIL$^+$ rate was also highest in the East zone (2.5%) and lowest in the South and West zones (1.8% and 2.0%, respectively p<.0001 for both comparisons). The South zone (where coverage and HSIL + rates were lower) had slightly higher estimated cancer rates compared with the East zone (with higher coverage and HSIL + rates), 38.1 (95% CI:27.8–50.0) vs. 48.5 (95% CI:37.4–61.8).

Among the 63 official neighborhoods of Manaus in 2019, the population size, the number of screening units and screening coverage varied widely (Fig 2). The estimated population per neighborhood ranged from 661 to 34,773 women age 25–64 (mean 8,003, median 5,089) (S2 Table). The most populated neighborhoods tended to have a higher number of women of screening age per screening unit and lower screening coverage. The number of women of screening age per screening unit varied between 420 to 7346 (mean 2,004, median 1,732) and

**Table 1. Estimated vaccination coverage for Manaus—Amazonas—Brazil, 2019.**

|       | Age group | Population[a] | Vaccinated[b] | | Coverage | |
|-------|-----------|---------------|---------------|----------|----------|----------|
|       |           |               | 1st dose | 2nd dose | 1st dose | 2nd dose |
| **Girls** | 9–13   | 109,175 | 80,120 | 51,772 | 73.4% | 47.4% |
|       | 14–18     | 111,910 | 96,171 | 79,815 | 85.9% | 71.3% |
| **Boys** | 10–13   | 89,304 | 38,684 | 19,810 | 43.3% | 22.2% |
|       | 14–15     | 44,139 | 28,638 | 14,691 | 64.9% | 33.3% |

[a] Population projections were provided by DICAR/SUBGS/SEMSA-Manaus;

[b]: SI-PNI (TABNET BD) (2019); Estimated number of vaccinations for the age cohort is extrapolated from annual age-stratified immunization totals.

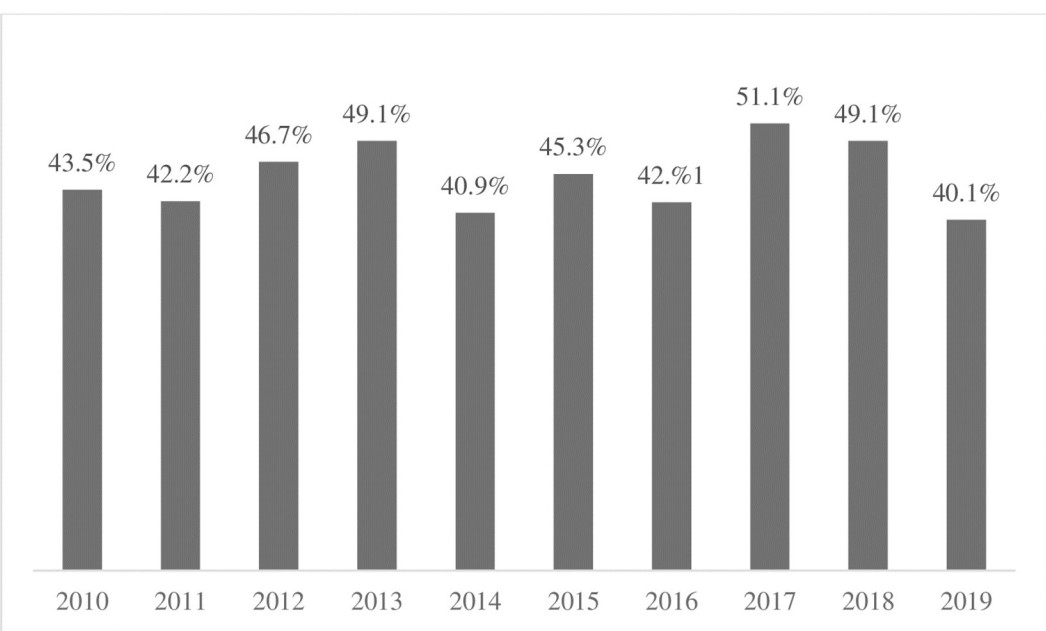

**Fig 1. Approximate annual screening coverage among women age 25–64, Manaus, Brazil, 2010–2019.** Approximate coverage = Number of cytology tests (screening and follow-up) processed at public laboratory among target population (1/3 of women age 25–64 in Manaus). Data sources: 2010 to 2013: SISCOLO / DATASUS; 2014 to 2016: SISCOLO-AM / DPCC (Local system). 2017 to 2018: SUS Outpatient Information System (SIA / SUS); 2019: Manaus Cytopathology Laboratory -SEMSA- Manaus (considering only Pap tests for screening).

was inversely correlated with screening coverage (mean 55.7%, range: 0–240%, median 44.0%) (Pearson's r = -0.39, p = .005).

During the 2019 Pink October campaign, we collaborated with 12 selected UBS´s to increase cervical cancer awareness and prioritize outreach to underscreened women, and 1,285 women age 25–64 were screened during the campaign (Table 3).

Overall, 70 (5.4%) women had HSIL$^+$ results requiring immediate referral to colposcopy. The HSIL$^+$ proportion varied greatly by UBS from 1.4% to 15.4%. Within 10 months after screening, follow-up information was found for 51 (72.8%) of the 70 women with HSIL$^+$; no

**Table 2. Screening program indicators among women aged 25–64, by urban residential zone of Manaus, Brazil. 2019.**

| Zone | Women population (age 25 to 64) | Cytology tests (only satisfatory samples) | Unsatisfatory cytology samples (%) | Annual coverage of target population (%) | Percent of cytology tests HSIL$^+$ (%) | Cervical cancers cases at reference cancer hospital (June-Dec 2019) | Annual number of cancers per 100,000 women age 25–64* (Confidence Interval) |
|---|---|---|---|---|---|---|---|
| South | 133,950 | 14,548 | 1.26 | 32.6 | 1.8 | 38 | 48.5 (37.4–61.8) |
| West | 128,532 | 14,272 | 1.38 | 33.3 | 2.0 | 32 | 42.7 (32.2–55.6) |
| North | 153,706 | 21,967 | 1.22 | 42.9 | 2.1 | 35 | 39.0 (29.7–50.2) |
| East | 120,818 | 19,779 | 1.43 | 49.1 | 2.5 | 27 | 38.1 (27.8–50.0) |
| **TOTAL** | **537,006** | **70,566** | **1.33** | **39.4** | **2.1** | **132** | |

p-value <.001 <.001.

HSIL$^+$: High-grade intraepithelial lesion or worse (includes atypical squamous cells high-grade (ASC-H, atypical glandular cells of undetermined significance (AGUS) and cancer). Screening coverage is number of screening cytologies among target population (1/3 of women age 25–64 in Manaus).

*Extrapolated from number of cases during 7-month period from June–December 2019. Cervical cancer includes only cases treated at FCECON. Chi-square p-value for trend.

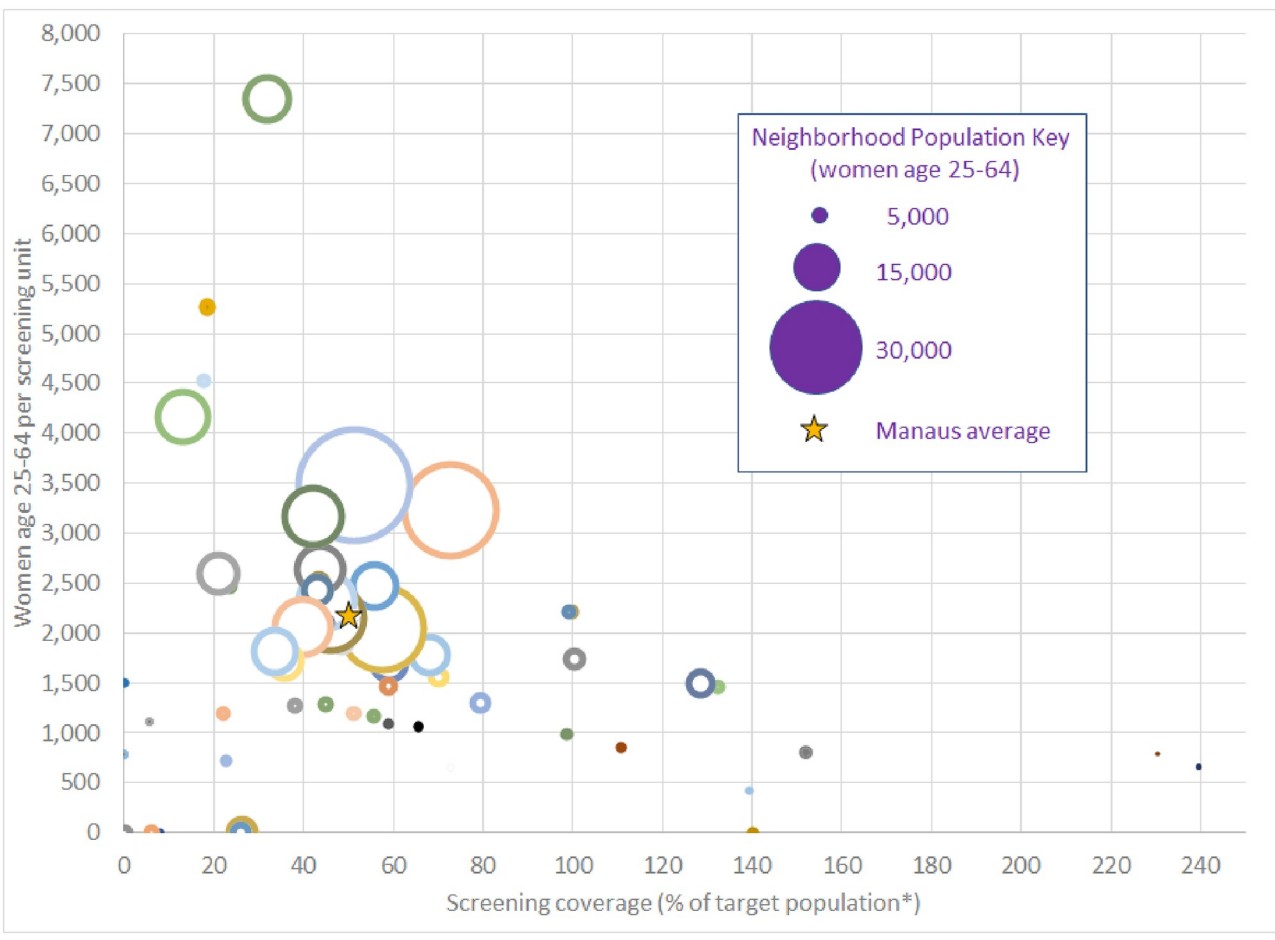

**Fig 2. Female population age 25–64, screening units, coverage per residential neighborhoods in urban Manaus, Brazil, 2019.** Estimated female population is represented by size of dots. Colors are used only to distinguish screening units. Dots on the x-axis indicate neighborhoods without screening units and their population uses units from nearby neighborhoods. *Some screening units attend to women who reside outside the area, at times resulting in coverage estimates of greater than 100% (particularly for smaller units).

evidence of follow-up was found for the other 27.1% (Fig 3). Among the 51 women attending colposcopy, 35 (68.6%) had a biopsy taken at the exam. Histopathology results were found for 23 women and showed: $6 \leq$ cervical intraepithelial neoplasia (CIN) grade 1, 15 CIN grade 2 or CIN grade 3, 1 squamous cell carcinoma (SCC) and 1 carcinosarcoma. Among 15 women diagnosed with CIN2 or CIN3, excisional treatment was confirmed for 10. The woman with a carcinosarcoma diagnosis died shortly after starting treatment. The woman with SCC received surgery, chemotherapy and radiotherapy. Biopsy and treatment were confirmed for 2 women with HSIL cytology, but no pathology results were found. As of 10 months after screening, we could not confirm complete work-up for 52 of 70 women with HSIL+ cytology.

## Discussion

Cervical cancer remains an important public health problem in many parts of Brazil, particularly lower resource areas such as the Amazonas and its capital Manaus [5]. In spite of a universal health care system with an expansive network of primary health centers, insufficient provision and utilization of preventive health services result in significant disparities in cancer

**Table 3. Screening tests and results among women attending selected basic health units, 2019 campaign (Manaus, Brazil).**

| Region/ Basic Health Unit | | Number of cytology tests 2019 campaign[a] | Total HSIL[+] results | |
| --- | --- | --- | --- | --- |
| | | | N | % |
| **South zone** | UBS 1 | 65 | 5 | 7.7 |
| | UBS 2 | 86 | 5 | 5.8 |
| | UBS 3 | 52 | 8 | 15.4 |
| | UBS 4 | 49 | 5 | 10.2 |
| **North zone** | UBS 1 | 151 | 7 | 4.6 |
| | UBS 2 | 120 | 5 | 4.2 |
| | UBS 3 | 98 | 3 | 3.1 |
| | UBS 4 | 211 | 3 | 1.4 |
| **East zone** | UBS 1 | 185 | 11 | 5.9 |
| | UBS 2 | 109 | 9 | 8.3 |
| | UBS 3 | 98 | 2 | 2.0 |
| | UBS 4 | 61 | 7 | 11.5 |
| **Total** | | **1,285** | **70** | **5.4** |

UBS: Unidade Básica de Saúde; HSIL[+]: High-grade intraepithelial lesion or worse (includes atypical squamous cells cannot rule out high grade (ASC-H), atypical glandular cells of undetermined significance (AGUS) and cancer).
[a]One-month screening campaign, Pink October, among women age 25–64.

burden [6]. Our review has identified critical challenges to screening programs in the city. Lower screening coverage and lower rates of abnormal screening results were correlated with elevated cervical cancer incidence, as observed in the south and west zones of urban Manaus. The yield of high-grade cytology results varied widely across city zones and health centers with better performance during the annual screening campaign. We observed cervical cancer screening was provided to many women outside the targeted age 25–64 years. Our review of

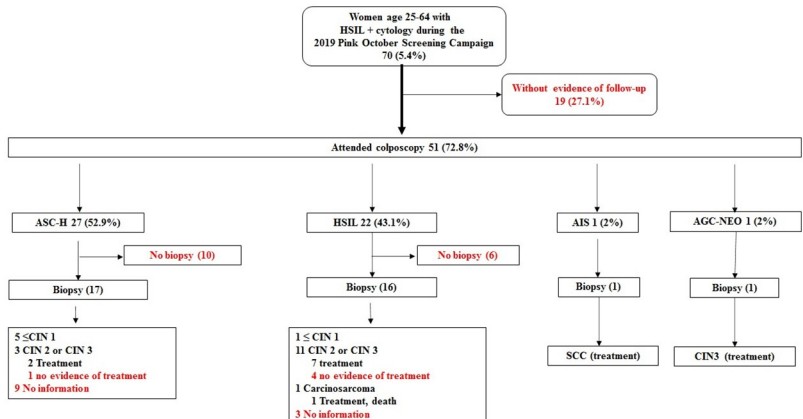

**Fig 3. Clinical follow-up and results for women with high-grade cytology result in Manaus, Brazil.** ASC-H: Atypical squamous cells, cannot rule-out high-grade squamous intraepithelial lesion; AGC: Atypical glandular cells; HSIL: High-grade squamous intraepithelial lesion; SCC: Squamous cell carcinoma. CIN1≤: Grade I Cervical Intraepithelial neoplasia, chronic cervicitis with squamous metaplasia and Ectocervix without significant histological changes. CIN2 ≥ (Grade II Cervical Intraepithelial Neoplasia, Grade III Cervical Intraepithelial Neoplasia). Months between cytology and colposcopy/biopsy: mean = 4.17 months, median = 2 months.

70 women with high-grade cytology results showed challenges attending colposcopy, taking biopsies, and receiving treatment; resulting in fewer than half of the women treated.

HPV vaccination coverage in the Amazonas has believed to be inadequate with declines after initial introduction in 2014 [17]. However, it is likely that calculations of annual coverage underestimate the overall vaccine protection because vaccinations over previous years were not aggregated [14]. Based upon public data from SUS, we estimate that the first cohorts of girls vaccinated (age 14–18 in 2019) have very high coverage, around 85% with one dose. The coverage for boys age 14–15 was lower around 64%. Vaccine coverage estimations are approximated and could be biased in either direction. No public central data system collects individual-level vaccination information, so we could only roughly estimate HPV vaccine coverage. In addition, the number of private sector HPV vaccinations were not considered here, and the population numbers are from the official 2010 census projections. Nonetheless, with such high vaccination rates among girls and in the absence of improved screening, in 50 years the cervical cancer rates in Manaus might be dramatically reduced since the HPV types in the quadrivalent vaccine are associated with 80% of cervical cancers. A forthcoming meta-analysis of type-distribution among Brazilian women will inform the potential local variability regarding vaccination impact.

The estimated coverage appears to be declining among girls, as in 2019, 73% of girls age 9–13 had one dose compared with 85% among older girls age 14–18. This concerning trend indicates the importance of continued programmatic vigilance in order to achieve sustained high coverage. This reduced trend coincides with the change of HPV immunization location from school-based in 2013 to clinic-based. To receive public assistance, documentation of full vaccination coverage including HPV is required, likely incentivizing a sizeable proportion of the population to vaccinate. But this strategy alone cannot achieve the high coverage observed with school-based programs. With the recent pandemic of SARS-CoV-2, some vaccination programs have nearly stopped. During this pause before programs are re-started and schools return to full in-class attendance, it might be worth considering whether HPV vaccination should be reintroduced into schools [18, 19]. This would be an appealing option given the observed social-cultural barriers [20] and logistical difficulties accessing health services [21]. Although girls and boys of all ages received fewer second doses, the accumulated evidence suggests that one-dose might confers sufficient protection [22–25] to reduce greatly the epidemic of HPV16 and HPV18 infection in young people. If one-dose strategies were to be implemented, vaccines could be available for catchup to a to-be-decided young adult age, in order to promote faster achievement of herd protection.

In our review of cervical cancers treated at the cancer hospital in Manaus, cases were observed across the city, both inside and outside catchment areas of health clinics, suggesting that the screening program was not functioning well, even for areas served by health clinics. Residence of cases was also inaccurate as some women indicate an address in an urban zone despite coming from rural areas or municipalities in the interior of Amazonas. Yet, despite these limitations, we observed an ecological correlation with less screening and lower HSIL$^+$ rates in zones with higher estimated cervical cancer rates. These observed disparities across city zones and even neighborhoods, are observed in other cities in Brazil and world-wide [8].

We estimated that approximately 40% of target female population (1/3 of women aged 25 to 64) had a cytology test in 2019, only half of the program's target of 80%. Of note, about one-third of the population in the urban area of Manaus uses private health services, including screening for cervical cancer. Even with the contribution of the private sector, screening coverage in Manaus remains low. Coverage might be even lower since the system does not allow us to identify women who are screened too frequently. Concerningly, one-quarter of the cytologies performed were among women outside the targeted screening age and thus unlikely to

benefit from screening. This is despite efforts by the Manaus Health Secretariat and the SUS Family Health Program to promote the Brazilian cervical cancer screening guidelines. Fortunately, Brazil has recently introduced performance targets for primary health care prevention services. In accordance with a new compensation model between the Brazilian Ministry of Health and local municipalities, a cytology performed on a woman outside the target age will no longer be reimbursed [26].

Cervical cancer screening programs are only effective if women at higher risk of cervical precancer are included in the screening program. Therefore, an elevated proportion of high-grade abnormalities usually indicates better coverage among high-risk women. In Brazil, the primary criteria used to assess risk and prioritize screening are age and HIV risk [11]. Yet, many studies worldwide have shown that the most important risk factor for cervical cancer is a woman's lack of quality screening [27]. Opportunistic screening programs in low-resource settings traditionally over-screen a minority of women while rarely, if ever, screening a significant proportion of women, typically those without access to health services [28]. Our analysis of the Manaus screening program depicts this scenario of inadequate use of limited resources. We found that in 2019, 2.1% of cytology tests conducted for screening were high-grade or worse (HSIL+), too few for a population with such a high cervical cancer incidence rate of 61.54 per 100,000 women [5]. Furthermore, when screening activities were closely monitored and teams were incentivized to increase participation during the 2019 Pink October campaign, the HSIL+ rates increased to 5.4% overall (ranging from 1.4 to 15.4% in selected UBS health centers). Among 70 women in select UBS health centers with HSIL+ cytology results, 54% had no record of having cytology in the past 3 years. We observed during the 2019 Pink October campaign that screening coverage in low resource settings can tangibly improve with training, capacity building, performing outreach activities and channeling efforts towards systematic screening.

Within the less-populated Rural Zone of Manaus a different scenario of screening coverage was observed. The geographically large zone covers 10,974 $km^2$ with a total female population aged 25 to 64 of only 2,807 in 2019 and consists of remote communities accessed by either land or river. The communities accessed by unpaved roads are covered by 4 UBS centers where cytology exams are performed daily. The 25 communities exclusively accessed by boat are visited every 20 days by one of two mobile floating UBS centers where cytology exams are performed. Local teams periodically (sometimes monthly) take cytology specimens to the Manaus public cytopathology laboratory for processing and to obtain results. The screening program is somewhat organized since the health teams map women's locations and community health agents are engaged to bring them for screening. In addition, this population relies on SUS for all health services and is strongly dependent upon government social programs which in turn require active participation in health and education programs. While the screening coverage appears to exceed program goals, it is likely falsely elevated due to frequent screening of women in the catchment area and screening women from surrounding areas outside Manaus. During 2019, 1,712 cytologies were performed among women in this age group, which represents, following the calculation of the indicator used by INCA, an apparent 177.8% coverage. On the other hand, the percentage of unsatisfactory samples was twice (2.20%) that observed in the urban area of Manaus (1.33%), although both percentages are within considered acceptable by the SUS quality system criterium of 5.0% or less [29]. In 2019, no cancer cases were reported from the Rural Zone. Importantly, women receiving follow-up diagnosis and treatment for precancer or cancer must temporary stay overnight in urban Manaus, resulting in possible inaccurate reporting of address and underreporting of cases within the Rural Zone.

In Manaus, for a woman who needs to undergo treatment for precursor lesions, the screening system design generally requires approximately 12 visits to health facilities (from the first screening visit to the day of the treatment itself). Until recently, screening has required three visits (one to make an appointment, another to attend screening and a third to retrieve the result 3 weeks later). Some clinics now offer limited walk-in screening without an appointment. Upon collection of her screening results, colposcopy is scheduled through the Regulation system (SISREG) for about 7 days later. Historically, if one or more biopsies were collected at colposcopy (as recommended for all high-grade cytology results), women had been responsible for taking their own biopsy samples to the SUS-contracted pathology laboratory. Biopsy delivery by SUS was initiated in 2019 but has been challenged by the COVID-19 pandemic. In 2019, the histopathology report used to be available in an average of 90 days and woman must collect her result from the laboratory to take to her follow-up visit where treatment is scheduled as needed.

Correspondingly, after extensive record review, we found that at that time, 70% of women with HSIL+ cytology results during the 2019 Pink October campaign at select UBS clinics had not completed the follow-up. Multiple reasons could contribute to this lack of follow-up including: never attended colposcopy, the provider did not take a biopsy (perhaps because no lesions were visualized), the provider did not receive histopathology results, or treatment was not yet completed. Anecdotally, we observed that for some women, their only follow-up was a repeat cytology. Our analysis is one of the few publications documenting individual-level colposcopy and treatment follow-up within SUS. We observed compliance far below a recent analysis of country-wide health services utilization data extrapolated that 93% of all colposcopies and 73% of required treatments were performed [30]. Due to the lack of a unified medical records system, follow-up activities are not linked nor tracked, making it difficult to identify procedures and necessitating extensive record review [19]. It is possible we did not capture all services, such as those provided by the private health sector. In addition, the SARS-CoV-2 epidemic hit Manaus about 5 months after 2019 Pink October screening campaign [31]. While follow-up visits should have been completed during this time, some delays might have been further observed due to the shutdown and distractions in the health system. Regardless, follow-up is a great challenge for women, local health clinic teams and Women's Health departments [32–35]. The newer SISCAN data system, released by SUS in 2013, is currently being introduced in the Manaus public health system and, especially if linked with other local SUS information systems will help alleviate this problem. However, UBS clinics have historically experienced challenges implementing such systems due to lack of computers and internet access.

It is evident that increased screening coverage alone does not translate to cervical cancer prevention without timely follow-up activities that result in treatment of precancer lesions. Strategies to improve compliance must address the educational and social challenges women experience through personalized attention by the health system. Other health systems in Brazil have shown that nursing teams, family health teams and community health workers can improve women's commitment with her own health and follow-up through intensive patient education and communication efforts [36–38]. In Manaus and other challenged areas of Brazil, screening programs would greatly benefit from removal of the many steps and delays that create obstacles, and thereby regain women's trust in and compliance with the screening program. For example, while participation of high-risk women was improved during the 2019 Pink October campaign in Manaus, follow-up remained a challenge.

New screening, triage and treatment methods are being used elsewhere to simplify and improve the effectiveness of screening programs with strategies that use more accurate tests, increase coverage and guarantee follow-up with adequate treatment [39–41]. The public

cytopathology lab in Manaus is well-organized and follows internationally-recognized quality control standards. However, HPV DNA testing has consistently been shown to be a more cost-effective approach compared with cytology in other settings because it is more sensitive and reproducible, allowing for extended screening intervals [42–44]. Furthermore, HPV DNA-based screening programs have the opportunity to use self-collection instead of clinician-collected strategies, which have been shown be accurate and widely acceptable, overcoming many barriers to reach underserved women [45, 46]. In a study in Amazonas state, Torres et al. found that self-collected HPV DNA testing was well-accepted by riverside women with low education and varying ages from isolated communities, indicating the promise of such a strategy in the Amazonian region [47].

HPV testing using a self-collection strategy is well-suited for high-volume screening campaigns and can even be conducted outside clinical settings, bringing screening closer to women in community locations [48, 49]. The logistics of sample transportation and processing in laboratories are much easier compared with cytology, allowing for quicker processing and provision of results. Molecular tests can be conducted from the same self-collected sample to inform how to triage HPV-positive women. Some HPV DNA tests identify the highest risk genotypes at the time of processing, thereby indicating those women who are at greatest risk of precancer and require referral to the health center. Methylation is a promising molecular triage test that, if proven feasible and cost-effective, could be useful for low resource settings [50]. The current standard-of-care method for triage of screen-positive women in Brazil is colposcopy, which has been shown world-wide to have multiple challenges [51]. Furthermore, we have found low compliance with the cumbersome processes for colposcopy and treatment. For many, but not all, cases a simplified safe treatment method such as thermal ablation could improve compliance, as endorsed by the WHO and proposed by specialists [52, 53]. With these new tools, a screening program could be envisioned that would include many fewer clinic visits for women, while improving quality of care.

We were able to obtain sufficient data to analyze multiple components of the cervical cancer prevention program in Manaus. But our analysis was limited by the inability to collect complete data at the woman-level regarding vaccination coverage, screening tests, or follow-up activities. In addition, services provided by the private sector, and non-registered cervical cancers were not available. Cervical cancer prevention programs require better informatics systems that can record vaccination as well as all steps of the screening and treatment strategy.

## Conclusions

In order to reduce incidence and mortality due to cervical cancer in Manaus, Amazonia region, and high-burden cities world-wide, public health system should assure high vaccination coverage through school-based or similar programs. Until the first cohort of vaccinated girls age past the peak of cervical cancer, secondary prevention will be needed to avert early deaths. For screening programs to be cost-effective, novel risk-based organized screening programs should use adaptable strategies to confront social determinants of health and reach high-risk women. Once identified, those at risk of developing cervical cancer must receive safe and effective treatment, with minimal hurdles for women and their providers.

## Supporting information

**S1 Table. Estimated 2019 demographic, geographic and health system organization of Manaus—Amazonas—Brazil.** Geographic and statistic Brazilian Institute (IBGE- 2019); \*\*Data from SIB/ANS/MS (http://www.ans.gov.br/anstabnet/cgi-bin/tabnet?dados/tabnet_tx. def)—12/2020 and population—DATASUS/MS—2012; \*\*\* Data from: SINANNET/GEVEP/

DEVAE (2016); [#] Data from 2017; [&] Data from PNUD/IPEA/FJP, 2010 and quote 1 American dollar = R$ 5.56.
(DOCX)

**S2 Table. Female population age 25–64, screening units, coverage per residential neighborhoods in urban Manaus, Brazil, 2019.** [*]A woman could have contributed more than one cytology. [#] Coverage in some neighborhoods was over and underestimated because women may use units from nearby neighborhoods.
(DOCX)

**S1 Fig. Geographical localization of Manaus (rural and urban area), Amazonas State, Brazil.** Source: adapted from IBGE cartographic base.
(TIF)

## Acknowledgments

We thank the technical teams of the Amazonas State Health Secretary (SES-AM), Manaus Municipal Health Secretary (SEMSA/MANAUS), Oncology Control Center Foundation (FCE-CON), COMPREV/INCA and PNI/MOH for their assistance obtaining and analyzing data.

## Author Contributions

**Conceptualization:** Kátia Luz Torres, Heidy Halanna de Melo Farah Rondon, Sandro Martins, Ana Ribeiro, Taina Raiol, Ana Cecilia Rodriguez, Julia C. Gage.

**Data curation:** Kátia Luz Torres, Heidy Halanna de Melo Farah Rondon, Toni Ricardo Martins, Sandro Martins, Ana Ribeiro, Taina Raiol, Ana Cecilia Rodriguez, Julia C. Gage.

**Formal analysis:** Ana Cecilia Rodriguez, Julia C. Gage.

**Investigation:** Kátia Luz Torres, Heidy Halanna de Melo Farah Rondon.

**Methodology:** Kátia Luz Torres, Heidy Halanna de Melo Farah Rondon, Toni Ricardo Martins, Sandro Martins, Ana Cecilia Rodriguez, Julia C. Gage.

**Project administration:** Kátia Luz Torres, Heidy Halanna de Melo Farah Rondon, Mark Schiffman, Ana Cecilia Rodriguez, Julia C. Gage.

**Supervision:** Kátia Luz Torres, Heidy Halanna de Melo Farah Rondon, Mark Schiffman, Ana Cecilia Rodriguez, Julia C. Gage.

**Validation:** Kátia Luz Torres, Heidy Halanna de Melo Farah Rondon, Toni Ricardo Martins, Sandro Martins, Ana Ribeiro, Taina Raiol, Carla Pintas Marques, Flavia Corrêa, Arn Migowski, Thais Tâmara Castro e Minuzzi-Souza, Mark Schiffman, Ana Cecilia Rodriguez, Julia C. Gage.

**Visualization:** Kátia Luz Torres, Heidy Halanna de Melo Farah Rondon, Ana Cecilia Rodriguez, Julia C. Gage.

**Writing – original draft:** Kátia Luz Torres, Heidy Halanna de Melo Farah Rondon, Ana Cecilia Rodriguez, Julia C. Gage.

**Writing – review & editing:** Kátia Luz Torres, Heidy Halanna de Melo Farah Rondon, Toni Ricardo Martins, Sandro Martins, Ana Ribeiro, Taina Raiol, Carla Pintas Marques, Flavia Corrêa, Arn Migowski, Thais Tâmara Castro e Minuzzi-Souza, Mark Schiffman, Ana Cecilia Rodriguez, Julia C. Gage.

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
