## [Decision Letter · Decision Letter 0]

9 Jun 2021

PONE-D-21-10794

Moving towards a strategy to accelerate cervical cancer elimination in a high-burden city – lessons learned from the Amazon city of Manaus, Brazil

PLOS ONE

Dear Dr. Torres,

Thank you for submitting your manuscript to PLOS ONE. After careful consideration, we feel that it has merit but does not fully meet PLOS ONE’s publication criteria as it currently stands. Therefore, we invite you to submit a revised version of the manuscript that addresses the points raised during the review process.

We look forward to receiving your revised manuscript.

Kind regards,

Luca Giannella

Academic Editor

PLOS ONE

Journal Requirements:

3. We note that Figure(s) S1 in your submission contain map images which may be copyrighted. All PLOS content is published under the Creative Commons Attribution License (CC BY 4.0), which means that the manuscript, images, and Supporting Information files will be freely available online, and any third party is permitted to access, download, copy, distribute, and use these materials in any way, even commercially, with proper attribution. For these reasons, we cannot publish previously copyrighted maps or satellite images created using proprietary data, such as Google software (Google Maps, Street View, and Earth). For more information, see our copyright guidelines: http://journals.plos.org/plosone/s/licenses-and-copyright.

a)  You may seek permission from the original copyright holder of Figure(s) S1 to publish the content specifically under the CC BY 4.0 license. 

4.Thank you for stating the following financial disclosure:

5.Thank you for including your ethics statement: "This research protocol was approved by the National Research Ethics Commission - CONEP, Brazil under number N. 3.676.252. The data were analyzed anonymously."

a) Please provide additional details regarding participant consent. In the ethics statement in the Methods and online submission information, please ensure that you have specified (1) whether consent was informed and (2) what type you obtained (for instance, written or verbal, and if verbal, how it was documented and witnessed). If your study included minors, state whether you obtained consent from parents or guardians. If the need for consent was waived by the ethics committee, please include this information.

Reviewers' comments:

Reviewer's Responses to Questions

**Comments to the Author**

1. Is the manuscript technically sound, and do the data support the conclusions?

Reviewer #1: Yes

Reviewer #2: Yes

Reviewer #3: Yes

Reviewer #4: Yes

Reviewer #5: Yes

2. Has the statistical analysis been performed appropriately and rigorously? 

Reviewer #1: Yes

Reviewer #2: Yes

Reviewer #3: Yes

Reviewer #4: Yes

Reviewer #5: Yes

3. Have the authors made all data underlying the findings in their manuscript fully available?

Reviewer #1: Yes

Reviewer #2: Yes

Reviewer #3: Yes

Reviewer #4: Yes

Reviewer #5: Yes

4. Is the manuscript presented in an intelligible fashion and written in standard English?

Reviewer #1: Yes

Reviewer #2: Yes

Reviewer #3: Yes

Reviewer #4: Yes

Reviewer #5: Yes

5. Review Comments to the Author

Reviewer #1: This is an excellent study of an important health issue.

The authors had pointed out the important limitation of the study regarding the difficulty in ascertaining if some tests are actually repeated for the same women.

Overall, very well conducted and written study.

Reviewer #2: Dear Authors,

I liked the manuscript emphasizing the importance of vaccination, screening and post-diagnostic followup for cervical lesions. Your desciption of the the potential gaps and offer for possible solutions can reduce incidence and mortality related to cervical cancer. This is a well written manuscript.

Regards,

Reviewer #3: The paper presents very important issues related to the prevention of cervical cancer. The selected research area correlates with the epidemiology of cervical cancer. The authors presented detailed information on the population covered by preventive measures. Based on the analysis of the obtained data, they formulated adequate and realistic conclusions.

Reviewer #4: Torres et al. present a manuscript on the realities of cervical cancer prevention in Manaus, Brazil. The study is based on a combination of a variety of public health care records which mostly cannot be followed through on the individual level. The manuscript describes both vaccination coverage of adolescents and the current cytology-based screening program.

The manuscript is descriptive of the current situation without opportunity for extensive statistical analyses or comparisons, but it still is an interesting read. In general, the manuscript is well written, and the few statistical analyses are performed adequately.

I have only minor comments/suggestions:

- I would prefer to see the cytopathology labotory(ies) further depicted (page 4 line 90-91) in Material and Methods; cytopathology laboratories are also mentioned in the Discussion (page 16 line 324 and page 17 line 360) – is there one centralized laboratory or several? Likewise, the number of clinics providing colposcopy services would be interesting to know.

- page 5 line 100; It would be interesting to know the number of unsatisfactory screening cytology samples as this also can describe the quality of the screening system and, also, if this varies between areas (as does the proportion of HSIL+ cytology samples)

- Figure 2 is a bit hard to evaluate as the meaning of different colors is not provided (are different colors necessary?), some further explanation of the figure would be nice. Further discussion of the outliers, those exceeding 100% coverage, could also be of value in the discussion (as is partially for the Rural Zone already). The * on the x-axis definition is not explained anywhere.

Reviewer #5: It is an interesting manuscript focused on the real world cervical cancer screening strategy in a high prevalence low resource reality in Brazil. Data is well presented and argumented and in the discussion section the most important topics are pointed out and properly discussed.

6. PLOS authors have the option to publish the peer review history of their article (what does this mean?). If published, this will include your full peer review and any attached files.

Reviewer #1: **Yes: **Nourah Al Qahtani

Reviewer #2: **Yes: **Funda Akpınar

Reviewer #3: No

Reviewer #4: No

Reviewer #5: **Yes: **Paolo Cristoforoni

---

## [Author Response · Author response to Decision Letter 0]

21 Aug 2021

July 19th, 2021

Dear Dr. Emily Chenette,

We present for your consideration the response to Reviewers for our research article manuscript entitled, “Moving towards a strategy to accelerate cervical cancer elimination in a high-burden city - lessons learned from the Amazon city of Manaus, Brazil”. 

Response to reviewers:

- We note that Figure(s) S1 in your submission contain map images which may be copyrighted. All PLOS content is published under the Creative Commons Attribution License (CC BY 4.0), which means that the manuscript, images, and Supporting Information files will be freely available online, and any third party is permitted to access, download, copy, distribute, and use these materials in any way, even commercially, with proper attribution. For these reasons, we cannot publish previously copyrighted maps or satellite images created using proprietary data, such as Google software (Google Maps, Street View, and Earth). 

o There is no copyrighted creation. The figure was constructed for the manuscript.

- Please clarify the sources of funding (financial or material support) for your study. 

o Financial disclosure: We have modified our financial disclosure statement in our cover letter with this new text: ‘This research was funded in part by the United States National Cancer Institute (NIH) Intramural Research Program which participated in the study design, analysis, decision to publish and preparation of the manuscript. The authors received no specific funding for this work.’ 

- Please provide additional details regarding participant consent. In the ethics statement in the Methods and online submission information, please ensure that you have specified (1) whether consent was informed and (2) what type you obtained (for instance, written or verbal, and if verbal, how it was documented and witnessed). If your study included minors, state whether you obtained consent from parents or guardians. If the need for consent was waived by the ethics committee, please include this information. 

o We amended the ethical statement in the Methods section of the manuscript (page 6 lines 127 to 129) and also added the same text to the “Ethics Statement” field in the submission form. The new text is: ‘Data were collected in collaboration with the local health authorities and the IRB waived the requirement for informed consent. For the analysis, all data were fully anonymized.’

- Minor comments/suggestions: I would prefer to see the cytopathology labotory(ies) further depicted (page 4 line 90-91) in Material and Methods; cytopathology laboratories are also mentioned in the Discussion (page 16 line 324 and page 17 line 360) – is there one centralized laboratory or several? Likewise, the number of clinics providing colposcopy services would be interesting to know. 

o We amended the introduction and methods sections providing information at page 4 line 69 and lines 89 to 91 to include more details regarding the number of colposcopy clinics and the unique centralized cytopathology laboratory, respectively.

- Minor comments/suggestions: page 5 line 100; It would be interesting to know the number of unsatisfactory screening cytology samples as this also can describe the quality of the screening system and, also, if this varies between areas (as does the proportion of HSIL+ cytology samples).

o We included a new column in table 2 with the percentage of Unsatisfactory cytology samples per urban area zones (page 8 line 172). We also discuss the unsatisfactory cytology percentage at urban and rural zones at page page 15 lines 314 and 317.

o We also included a new reference, number 29, regarding the Brazilian parameter for unsatisfatory cytology samples.

- Minor comments/suggestions: Figure 2 is a bit hard to evaluate as the meaning of different colors is not provided (are different colors necessary?), some further explanation of the figure would be nice. Further discussion of the outliers, those exceeding 100% coverage, could also be of value in the discussion (as is partially for the Rural Zone already). The * on the x-axis definition is not explained anywhere.

o We amended Fig 2 footnote to clarify the meaning of the different colors and the asterisk. The new text, page 9 lines 188 to 192. “Estimated female population is represented by size of dots. Colors are used only to distinguish screening units. Dots on the x-axis indicate neighborhoods without screening units whose population attends screening units from nearby neighborhoods. *Some screening units attend to women who reside outside the area, at times resulting in coverage estimates of greater than 100% (particularly for smaller units).”

o Regarding the outliers, we discuss the potential explanation in the discussion section page 15 lines 311 to 313. 

We adjusted the text of reference citation number 13 as follows: Instituto Brasileiro de Geografia e Estatísticas IBGE [Internet]. 2019. Available from: https://cidades.ibge.gov.br/brasil/am/manaus/panorama.

---

## [Decision Letter · Decision Letter 1]

30 Sep 2021

Moving towards a strategy to accelerate cervical cancer elimination in a high-burden city – lessons learned from the Amazon city of Manaus, Brazil

PONE-D-21-10794R1

Dear Dr. Torres,

We’re pleased to inform you that your manuscript has been judged scientifically suitable for publication and will be formally accepted for publication once it meets all outstanding technical requirements.

Kind regards,

Luca Giannella

Academic Editor

PLOS ONE

Additional Editor Comments (optional):

Reviewers' comments:

Reviewer's Responses to Questions

**Comments to the Author**

1. If the authors have adequately addressed your comments raised in a previous round of review and you feel that this manuscript is now acceptable for publication, you may indicate that here to bypass the “Comments to the Author” section, enter your conflict of interest statement in the “Confidential to Editor” section, and submit your "Accept" recommendation.

Reviewer #4: All comments have been addressed

2. Is the manuscript technically sound, and do the data support the conclusions?

Reviewer #4: Yes

3. Has the statistical analysis been performed appropriately and rigorously? 

Reviewer #4: Yes

4. Have the authors made all data underlying the findings in their manuscript fully available?

Reviewer #4: Yes

5. Is the manuscript presented in an intelligible fashion and written in standard English?

Reviewer #4: Yes

6. Review Comments to the Author

Reviewer #4: (No Response)

7. PLOS authors have the option to publish the peer review history of their article (what does this mean?). If published, this will include your full peer review and any attached files.

Reviewer #4: No

---

## [Editor Report · Acceptance letter]

7 Oct 2021

PONE-D-21-10794R1 

Moving towards a strategy to accelerate cervical cancer elimination in a high-burden city – lessons learned from the Amazon city of Manaus, Brazil. 

Dear Dr. Torres:

I'm pleased to inform you that your manuscript has been deemed suitable for publication in PLOS ONE. Congratulations! Your manuscript is now with our production department. 

Kind regards, 

on behalf of

Dr. Luca Giannella 

Academic Editor

PLOS ONE